Semantic micro-contributions with decentralized nanopublication services

http://orcid.org/0000-0002-1267-0234 Kuhn Tobias 1 t.kuhn@vu.nl
Taelman Ruben 2
http://orcid.org/0000-0002-1501-1082 Emonet Vincent 3
Antonatos Haris 4
http://orcid.org/0000-0001-9842-9718 Soiland-Reyes Stian 5 6
http://orcid.org/0000-0003-4727-9435 Dumontier Michel 3
1 Department of Computer Science, VU Amsterdam , Amsterdam , Netherlands
2 IDLab, Ghent University , Ghent , Belgium
3 Institute of Data Science, Maastricht University , Maastricht , Netherlands
4 SciFY , Athens , Greece
5 Informatics Institute, University of Amsterdam , Amsterdam , Netherlands
6 Department of Computer Science, The University of Manchester , Manchester , UK
Ghidini Chiara
Electronic publication date: 2021 Mar 8
Publication date: 2021
Volume: 7
Electronic Location ID: e387
Received 2020 Aug 26; Accepted 2021 Jan 19
Copyright: © 2021 Kuhn et al.
Copyright year: 2021
Copyright holder: Kuhn et al.
License: This is an open access article distributed under the terms of the Creative Commons Attribution License, which permits unrestricted use, distribution, reproduction and adaptation in any medium and for any purpose provided that it is properly attributed. For attribution, the original author(s), title, publication source (PeerJ Computer Science) and either DOI or URL of the article must be cited.
License URL: https://creativecommons.org/licenses/by/4.0/

Keywords: Nanopublications, Semantic Web, Linked data, Semantic publishing

Funding: Research Foundation—Flanders (FWO) 1274521N National Institutes of Health OT2TR003434-01 BioExcel-2 (European Commission) H2020-INFRAEDI-02-2018-823830 Ruben Taelman is a postdoctoral fellow of the Research Foundation — Flanders (FWO) (1274521N). Support for Vincent Emonet and Michel Dumontier was provided by the Biomedical Data Translator project funded by National Institutes of Health (No. OT2TR003434-01). Stian Soiland-Reyes was funded by BioExcel-2 (European Commission H2020-INFRAEDI-02-2018-823830). The funders had no role in study design, data collection and analysis, decision to publish, or preparation of the manuscript.

==============================
While the publication of Linked Data has become increasingly common, the process tends to be a relatively complicated and heavy-weight one. Linked Data is typically published by centralized entities in the form of larger dataset releases, which has the downside that there is a central bottleneck in the form of the organization or individual responsible for the releases. Moreover, certain kinds of data entries, in particular those with subjective or original content, currently do not fit into any existing dataset and are therefore more difficult to publish. To address these problems, we present here an approach to use nanopublications and a decentralized network of services to allow users to directly publish small Linked Data statements through a simple and user-friendly interface, called Nanobench, powered by semantic templates that are themselves published as nanopublications. The published nanopublications are cryptographically verifiable and can be queried through a redundant and decentralized network of services, based on the grlc API generator and a new quad extension of Triple Pattern Fragments. We show here that these two kinds of services are complementary and together allow us to query nanopublications in a reliable and efficient manner. We also show that Nanobench makes it indeed very easy for users to publish Linked Data statements, even for those who have no prior experience in Linked Data publishing.

Introduction

Linked Data has achieved remarkable adoption (Bizer, Heath & Berners-Lee, 2011; Schmachtenberg, Bizer & Paulheim, 2014), but its publication has remained a complicated issue. The most popular methods for publishing Linked Data include subject pages (Berners-Lee, 2009), SPARQL endpoints (Feigenbaum et al., 2013), and data dumps. The latter are essentially just RDF files on the web. Such files are not regularly indexed on a global scale by any of the existing search engines and therefore often lack discoverability, but they are the only option that does not require the setup of a web server for users wanting to publish Linked Data on their own. While one of the fundamental ideas behind the web is that anyone should be able to express themselves, Linked Data publishing is therefore mostly done by large centralized entities such as DBpedia (Auer et al., 2007) and Wikidata (Vrandečić & Krötzsch, 2014). Even such community-driven datasets have clear guidelines on what kind of data may be added and typically do not allow for subjective or original content, such as personal opinions or new scientific findings that have otherwise not yet been published. It is therefore difficult for web users to publish their own personal pieces of Linked Data in a manner that the published data can be easily discovered, queried, and aggregated. To solve these shortcomings, we propose here a complementary approach to allow for what we call semantic micro-contributions. In contrast to the existing Linked Data publishing paradigms, semantic micro-contributions allow individual web users to easily and independently publish small snippets of Linked Data. We show below how such semantic micro-contributions can be achieved with nanopublications and semantic templates, and how we can make such a system redundant and reliable with a decentralized network of services. We will explain below how this approach differs from other decentralization approaches that have been proposed in the context of Linked Data publishing (including Solid and Blockchain-based approaches).

Concretely, we investigate here the research question of how we can build upon the existing nanopublication publishing ecosystem to provide query services and intuitive user interfaces that allow for quick and easy publishing of small Linked Data contributions in a decentralized fashion. Our concrete contributions are:a concrete scheme of how nanopublications can be digitally signed and thereby reliably linked to user identities,

two complementary sets of nanopublication query services building upon extensions of existing Linked Data technologies, one based on the grlc API generator and the other one in the form of an extension of Triple Pattern Fragments called Quad Pattern Fragments (QPF),

a user interface connecting to these services that allows for simple nanopublication publishing based on the new concept of nanopublication templates, and

positive evaluation results on the above-mentioned query services and user interface.

Below, we outline the relevant background, introduce the details of our approach, present and discuss the design and results of two evaluations, and outline future work.

Background

Before we introduce our approach, we give here the relevant background in terms of our own previous work, and other related research on the topics of the use of semantic technologies for scientific publishing, Linked Data APIs, and decentralization.

Under the label of semantic publishing (Shotton, 2009), a number of approaches have been presented to align research and its outcomes with Linked Data in order to better organize, aggregate, and interpret scientific findings and science as a whole. We have previously argued that these Linked Data representations should ideally come directly from the authors (i.e., the researchers), should cover not just metadata properties but the content of the scientific findings themselves, and should become the main publication object instead of papers with narrative text, in what we called genuine semantic publishing (Kuhn & Dumontier, 2017). Nanopublications (Mons et al., 2011) are one of the most prominent proposals to implement this. They are small independent pieces of Linked Data that encapsulate atomic statements in the form of a few RDF triples (this part is called the assertion graph) together with formal provenance information (the provenance graph, e.g., pointing to the study that the assertion was derived from) and metadata (the publication info graph, e.g., by whom and when was the nanopublication created). While the original nanopublication proposal focused on assertions with domain statements (such as expressing a link between a gene and a disease), we subsequently suggested to broaden their scope and to use them also to express bibliographic and other meta-level information, statements about other nanopublications, vocabulary definitions, and generally any kind of small and coherent snippet of Linked Data (Kuhn et al., 2013). In order to make nanopublications verifiable and to enforce their immutability, we then showed how cryptographic hash values can be calculated on their content and included in their identifiers in the form of trusty URIs (Kuhn & Dumontier, 2015). Based on this, we established a decentralized and open server network, through which anybody can reliably publish and retrieve nanopublications (Kuhn et al., 2016), and we introduced index nanopublications, which allow for assigning nanopublications to versions of larger datasets (Kuhn et al., 2017). The work to be presented below is a continuation of this research line, adding query services and an intuitive publishing interface as components to this ecosystem.

Our general approach is partly related to semantic wikis, for example, Ghidini et al. (2009), Baumeister, Reutelshoefer & Puppe (2011) and Kuhn (2008). They combine the ideas of the Semantic Web with the wiki concept, and therefore allow for quick and easy editing of semantic data. They focus on the collaborative process of consensus finding and its result in the form of a single coherent formal knowledge base, and as such, they focus less on individual contributions as the unit of reference.

In terms of Linked Data APIs, SPARQL endpoints (Feigenbaum et al., 2013) are probably the most well-known example and they are often used for providing queryable access to RDF datasets. In practice, such endpoints often suffer from availability problems (Buil-Aranda et al., 2013), due to their public nature and the uncontrolled complexity of SPARQL queries. The Linked Data Fragments (LDF) framework (Verborgh et al., 2016) was initiated as an attempt to investigate alternative RDF query interfaces, where the total query effort can be distributed between server and client. Triple Pattern Fragments (TPF) (Verborgh et al., 2016), for example, heavily reduce the expressivity of queries that can be evaluated by a server, so clients that want answers to more complex SPARQL queries need to take up part of the execution load themselves. Through client-side query engines, such as Comunica (Taelman et al., 2018), complex SPARQL queries can be split into multiple triple pattern queries that can be executed separately by a TPF server and then joined to create the full result on the client-side. Another approach to address the problems of full SPARQL endpoints is grlc (Meroño-Peñuela & Hoekstra, 2016), a tool that automatically generates APIs from SPARQL templates. By providing a small number of possible API operations instead of SPARQL’s virtually unlimited query possibilities, grlc makes Linked Data access easier and better manageable on both, the client and server side. A further noteworthy technology is the Linked Data Platform (LDP) (Speicher, Arwe & Malhotra, 2015) to manage and provide Linked Data. In order to establish connections between producers and consumers of Linked Data, subscription and notification protocols such as WebSub (https://www.w3.org/TR/websub/) and provenance pingbacks (https://www.w3.org/TR/prov-aq/#provenance-pingback) have been proposed.

The approaches above mostly assume requests are targeted towards a central server. This centralization comes with the downsides that such a server forms a single point of failure, that we need to trust in the authority that runs it, and that it is difficult to scale. To address these problems, a number of more decentralized approaches have been proposed. LDF interfaces such as TPF, as introduced above, can in fact also be used in a more distributed fashion, as fragments can be published across different servers (Delva et al., 2019). Distributed approaches to semantically annotate web pages like https://schema.org/ (Guha, Brickley & Macbeth, 2016) have moreover shown strong adoption. Another example is Solid (Mansour et al., 2016), where users have their own personal Linked Data pod, in which they can store their own data and thereby are in full control of who can access it. Solid thereby targets personal and potentially confidential data, with a focus on access control and minimizing data duplication. The Solid ecosystem has been applied in a number of use cases, such as collaboration within decentralized construction projects (Werbrouck et al., 2020), and decentralization of citizen data within governments (Buyle et al., 2019). Such approaches where data is distributed but not replicated, however, often lead to major difficulties when queries need to be executed over such a federation of data sources (Taelman, Steyskal & Kirrane, 2020).

This stands in contrast to decentralized approaches where data is not only distributed but also replicated, which typically target open and public data and have an emphasis on scalability and reliability. Blockchain-based solutions fall into the latter category, for which a whole range exists of possible approaches to integrate Linked Data (Third & Domingue, 2017). A core trade-off of all blockchain-based approaches is to either sacrifice some degree of decentralization with permissioned blockchains or to pay the price of the expensive mining process. For applications that do not crucially depend on a fixed and agreed-upon order of events, as cryptocurrencies do for their transaction ledger, the costs of Blockchain-based solutions in fact often do not seem to offset their benefits. Our approach to be presented below also falls into this second category of decentralization approaches with replicated data sources, but does not entail the costs of Blockchain-based approaches.

Approach

The approach to be presented here, as shown in Fig. 1, is based on our work on nanopublications and the ecosystem for publishing them, as introduced above. The core of this new approach is to allow end-users to directly publish Linked Data snippets in the form of nanopublications with our existing decentralized nanopublication publishing network through an interface powered by semantic templates, which are themselves published as nanopublications. Below we explain how users can establish their identity by announcing their public key, and how they can then sign, publish, and update their nanopublications. Then we describe our extension of Triple Pattern Fragments to support quads and thereby nanopublications. Next, we show how we defined two complementary sets of services on top of the existing nanopublication network to query and access the published data in a redundant and reliable way. Finally, we explain how these components together with semantic templates allowed us to build a flexible and intuitive end-user application called Nanobench.

Figure 1 The architecture of our overall approach.

Identities and updates

Nanopublications typically specify their creator in the publication info graph, but because anybody can publish anything they want through the existing open nanopublication network, there is no guarantee that this creator link is accurate. For that reason, we propose here a method to add a digital signature to the publication graph. With our approach, users have to first introduce their identifier and public key before they can publish their own nanopublications. This introduction is itself published as a signed nanopublication declaring the link between the personal identifier (such as an ORCID identifier) and the public key in its assertion graph, as shown by this example:

sub:assertion {

sub:keyDeclaration npx:declaredBy orcid:0000-0001-2345-6789 ;

npx:hasAlgorithm "RSA";

npx:hasPublicKey "MIGfMA0GCSqGSIb3DQEBAQUAA4GNADCBiQK…" .

}

Below, we will come back to the question of how we can ensure that this user is indeed in control of the stated ORCID identifier. Once an identity is established in this way, the respective user can publish nanopublications such as the one shown in Fig. 2, where the personal identifier and the public key are mentioned in the publication info graph (yellow) together with the digital signature that is calculated with the respective private key on the entire nanopublication, excluding only the npx:hasSignature triple and the hash code of the trusty URI. The trusty URI (here represented with the prefix this: ) is calculated as a last step, which therefore also covers the signature. This makes the nanopublication including its signature verifiable and immutable.

Figure 2 Example nanopublication in TriG notation that was published with Nanobench.

Immutability is a desirable property to ensure stable and reliable linking, but for practical purposes it has to come with a mechanism to declare updates and mark obsolete entries. With our approach, new versions of a nanopublication can be declared with the npx:supersedes property in the publication info graph of the nanopublication containing the update, for example:

sub:pubinfo {

this: npx:supersedes

<http://purl.org/np/RAvjbXCGsF1R03yUjeAHC2arCGqTtn5BThOEkz4HPfPrc> .

…

}

In order to declare a nanopublication obsolete without an update, the npx:retracts property can be used in the assertion graph of a separate retraction nanopublication, for example:

sub:assertion {

orcid:0000-0001-2345-6789 npx:retracts

<http://purl.org/np/RALS50Z57WzbjVsj2mZLAIX34_GicNnn2RMAlZd-yjpYo> .

}

Of course, updated versions and retractions should only be considered valid if authorized by the author of the original nanopublication. For the scope of this work, we only consider them valid if the retraction or update is signed with the same key pair, but more flexible solutions are possible in the future.

The elements introduced so far allow us to cryptographically verify that given nanopublications were published by the same user who introduced herself in her introduction nanopublication, but they still allow anybody to claim any ORCID identifier (or other kind of identifier). To add this missing link, users can add the link of their introduction nanopublication to their ORCID profile under “Websites & Social Links”, which proves that they have control of that account. This link is represented with foaf:page when the user identifier is resolved with a HTTP GET request asking for an RDF representation via content negotiation. This is thereby a general method that can work on any URL scheme and identification mechanism providing dereferenceable user identifiers, but for simplicity we will restrict our discussion here to ORCID identifiers.

Quad pattern fragments

Nanopublications, as can be seen in Fig. 2, are represented as four named RDF graphs. Triple Pattern Fragments, however, as their names indicates, only support triples and not quads (which include the graph information), and TPF is therefore insufficient for querying nanopublications. For this reason, we introduce an extension of TPF to support quads, called Quad Pattern Fragments (QPF) (https://linkeddatafragments.org/specification/quad-pattern-fragments/).

In order to allow querying over QPF, its HTTP responses include metadata that declaratively describe the controls via which querying is possible. These controls are defined in a similar way as for TPF using the Hydra Core vocabulary (Lanthaler & Gütl, 2013), and allows intelligent query engines to detect and use them. Below, an example of these controls is shown:

@prefix rdf: <http://www.w3.org/1999/02/22-rdf-syntax-ns#>.

@prefix hydra: <http://www.w3.org/ns/hydra/core#>.

@prefix void: <http://rdfs.org/ns/void#>.

@prefix sd: <http://www.w3.org/TR/sparql11-service-description/#>.

<https://example.org/#dataset> a void:Dataset, hydra:Collection;

void:subset <https://example.org/>;

sd:defaultGraph <urn:ldf:defaultGraph>;

hydra:search _:pattern.

_:pattern hydra:template "https://example.org/{?s,p,o,g}";

hydra:variableRepresentation hydra:ExplicitRepresentation;

hydra:mapping _:subject, _:predicate, _:object, _:graph.

_:subject hydra:variable "s";

hydra:property rdf:subject.

_:predicate hydra:variable "p";

hydra:property rdf:predicate.

_:object hydra:variable "o";

hydra:property rdf:object.

_:graph hydra:variable "g";

hydra:property sd:graph.

The control above indicates that the QPF API accepts four URL parameters, corresponding to the four elements of a quad. For example, a query to this API for the pattern ?s npx:retracts ?o sub:assertion would result in an HTTP request for the URL https://example.org/?p=npx:retracts&g=sub:assertion1 .

Just like with TPF, intelligent clients can be built that can handle more complex queries (such as SPARQL queries) over QPF APIs. This requires these clients to split up a SPARQL query into multiple quad patterns, which can be resolved by the API, after which they can be joined by the client to form a complete query result.

QPF has been designed to be backwards-compatible with TPF. This means that clients that implement support for TPF APIs, but do not understand the notion of QPF, will be able to recognize the API as TPF, and execute triple pattern queries against it. Due to the declaratively described QPF and TPF controls, clients such as the Comunica engine can recognize and make use of both variants next to each other. A live version of a QPF API can be found at https://ldf.nanopubs.knows.idlab.ugent.be/np, which is one of six instances of this service in our network2 .

Nanopublication services

Nanopublications can be reliably and redundantly published by uploading them to the existing nanopublication server network (Kuhn et al., 2016), which at the time of writing consists of eleven severs in five countries and storing more than 10 million nanopublications (http://purl.org/nanopub/monitor). This network implements a basic publishing layer where nanopublications can be looked up by their trusty URI, but no querying features are provided.

In order to allow for querying of the nanopublications’ content, we present here our implementation of a new service layer built on top of the existing publication layer. While we are using a triple store with SPARQL under the hood, we do not provide a full-blown SPARQL endpoint to users in order to address the above-mentioned problems of availability and scalability. For our nanopublication service layer, we employ a mix of two kinds of services that are more restricted than SPARQL but also more scalable. The first kind of service is based on LDF via our QPF API, as introduced above, and allows only for simple queries at the level of individual RDF statements but does not impose further restrictions. The second one is based on the grlc API generator (Meroño-Peñuela & Hoekstra, 2016), which optionally comes with the Tapas HTML interface (Lisena et al., 2019) and which can be used to execute complex queries but is restricted to a small number of predefined patterns.

The LDF-based services reduce the complexity and load on the server by only allowing for very simple queries to be asked to the server, and delegate the responsibility of orchestrating them to answer more complex questions to the client. The grlc-based services reduce the complexity and load by allowing only for queries that are based on a small number of SPARQL templates that are hand-crafted for optimized performance. These two kinds of services are thereby designed to be complementary, with grlc being restricted but faster and LDF being more powerful but slower.

The grlc-based services provide general API operations that are based on 14 SPARQL templates:find_nanopubs returns all nanopublication identifiers in undefined order (paginated in groups of 1,000) possibly restricted by the year, month, or day of creation;

find_nanopubs_with_pattern additionally allows for specifying the subject, predicate, and/or object of a triple in the nanopublication as a filter, and to restrict the occurrence of that triple to the assertion, provenance, or publication info graph;

find_nanopubs_with_uri similarly allows for filtering by a URI irrespective of its triple position;

find_nanopubs_with_text supports full-text search on the literals in the nanopublication (using non-standard SPARQL features available in Virtuoso and GraphDB);

for each of the four find_nanopubs_* templates mentioned above, there is also a find_signed_nanopubs_* version that only returns nanopublications that have a valid signature and that allows for filtering by public key;

get_all_indexes returns all nanopublication indexes (i.e., sets of nanopublications);

get_all_users returns all users who announced a public key via an introduction nanopublication;

get_backlinks returns all identifiers of nanopublications that directly point to a given nanopublication;

get_deep_backlinks does the same thing but includes deep links through chains of nanopublications linking to the given one;

get_latest_version returns the latest version of a given nanopublication signed by the same public key by following npx:supersedes backlinks;

get_nanopub_count returns the number of nanopublications, possibly restricted by year, month, or day of creation.

The full SPARQL templates can be found in the Supplemental Material (see below). These API calls provide a general set of queries based on which applications with more complex behavior can be built. We will introduce Nanobench as an example of such an application below.

In order to answer some of the above queries, auxiliary data structures have to be created while loading new nanopublications. Most importantly, digital signatures cannot be checked in SPARQL directly, as this involves translating the triples of a nanopublication into a normalized serialization and then calculating a cryptographic hash function on it, which goes beyond SPARQL’s capabilities. Other aspects like deep backlinks are complicated because it is not sufficient to check whether a link is present, but we also need to check that the respective triple is located in the linking nanopublication (as a triple linking two nanopublications could itself be located in a third nanopublication). In order to solve these problems, additional triples in two administrative graphs are generated when new nanopublications are loaded. Concretely, the following triples are added for each nanopublication (placeholders in capitals):

npa:graph {

<NPURI> npa:hasHeadGraph <HEADURI> ;

dct:created "DATETIME"^^xsd:dateTime ;

npa:creationDay <http://purl.org/nanopub/admin/date/YEAR-MONTH-DAY> ;

npa:creationMonth <http://purl.org/nanopub/admin/date/YEAR-MONTH> ;

npa:creationYear <http://purl.org/nanopub/admin/date/YEAR> ;

npa:hasValidSignatureForPublicKey "PUBLICKEY" .

}

npa:networkGraph {

<NPURI> <INTER-NP-PREDICATE> REFERENCED-NPURIS… .

<NPURI> npa:refersToNanopub REFERENCED-NPURIS… .

}

The first triple of the npa:graph links the nanopublication identifier to its head graph, where the links to its assertion, provenance, and publication info graphs can be found. The second one contains the creation date in a normalized form. Number three to five allow for efficient filtering by day, month, and year, respectively (we use URIs instead of literals because this happens to be much faster for filtering under Virtuoso). The final triple in the npa:graph links the nanopublication to its public key if the signature was found to be valid.

In the npa:networkGraph, all instances of linking to another nanopublication with the linking nanopublication URI in subject position are added (e.g., with npx:supersedes). In the cases where another nanopublication is linked but not with the pattern of the linking nanopublication in subject position (e.g., as with npx:retracts), npa:refersToNanopub is used as predicate to link the two nanopublications.

We set up a network of six servers in five different countries each providing both of the introduced services (LDF-based and grlc-based). They are notified about new nanopublications by the servers of the existing publishing network, which are otherwise running independently. The services connect to a local instance of a Virtuoso triple store (https://virtuoso.openlinksw.com/), into which all nanopublications are loaded via a connector module. This connector module also creates the additional triples in the administrative graphs as explained above.

While the restriction to predefined templates with grlc significantly improves the scalability of the system as compared to unrestricted SPARQL, further measures will be needed in the future if the number of nanopublications keeps growing to new orders of magnitude. The services presented here are designed in such a way that such measures are possible with minimal changes to the API. The 14 query templates of the grlc services can be distributed to different servers, for example, such that a single server would only be responsible for one of the 14 kinds of queries. This server could then use an optimized data structure for exactly that kind of query and would only need to hold a fraction of the data. The find_ queries could moreover be further compartmentalized based on publication date, for example each server instance just covering a single year. The LDF-based services could be distributed in a similar fashion, for example based on the predicate namespace.

Nanobench client and templates

To demonstrate and evaluate our approach, we next implemented a client application that runs on the user’s local computer, can be accessed through their web browser, and connects to the above decentralized network of services. The code can be found online (https://github.com/peta-pico/nanobench) and Fig. 3 shows a screenshot.

Figure 3 A screenshot of the Nanobench application with a publication form.

In the “search” part of the interface, users are provided with a simple search interface that connects to the grlc API operations find_nanopubs_with_uri (if a URI is entered in the search field) or find_nanopubs_with_text (otherwise). In the “others” part, other users’ latest nanopublications can be seen in a feed-like manner, similar to Twitter feeds.

In order for users to publish their own nanopublications and thereby create their own feed, they have to first set up their profile. Nanobench provides close guidance through this process, which involves the declaration of the user’s ORCID identifier, the creation of an RSA key pair, and the publication of an introduction nanopublication that links the public key to the ORCID identifier. The last step of linking the new introduction nanopublication from the user’s ORCID profile is not strictly necessary for the user to start publishing nanopublications and is therefore marked as optional.

Once the user profile is completed, a list of templates is shown in the “publish” part of the interface. Templates are published as nanopublications as well, and so this list can be populated via a call to the find_signed_nanopubs_with_pattern operation of the grlc-based services. Currently, the list includes templates for free-text commenting on a URL, expressing a foaf:knows relation to another person, declaring that the user has read a given paper, expressing a gene–disease association, retracting a nanopublication, describing a datasets with a SPARQL endpoint, and publishing an arbitrary RDF triple. After selecting a template, a form is automatically generated that allows the user to fill in information according to that template, as shown in Fig. 3.

Templates describe the kind of statements users can publish and also provide additional information on how the input form should be presented to the user. This is an example of a template (the same one that is shown in Fig. 3), defined in the assertion graph of a nanopublication:

sub:assertion {

sub:assertion a nt:AssertionTemplate ;

rdfs:label "Expressing that you know somebody" ;

nt:hasStatement sub:st1 .

sub:st1 a rdf:Statement ;

rdf:subject nt:CREATOR ;

rdf:predicate foaf:knows ;

rdf:object sub:person .

foaf:knows rdfs:label "know" .

sub:person a nt:UriPlaceholder ;

rdfs:label "ORCID identifier of the person you know" ;

nt:hasPrefix "https://orcid.org/" ;

nt:hasRegex "[0-9]{4}-[0-9]{4}-[0-9]{4}-[0-9]{3}[0-9X]" .

}

In a template nanopublication, the assertion graph is classified as an AssertionTemplate (in the namespace https://w3id.org/np/o/ntemplate/) and given a human readable label with rdfs:label. Moreover, it is linked to the statement templates (i.e., triples in the nanopublications to be published) via hasStatement. The above example has just one such statement template, but more complex templates involve several of them. These templates then use regular RDF reification to point to their subjects, predicates, and objects. In the case of multiple statements, their order in the form can be defined with statementOrder and some of them can be marked as optional by classifying them as OptionalStatement. rdfs:label can be used on all the elements to define how they should be labeled in the form interface, and the special URI CREATOR is mapped to the identifier of the user applying the template.

Importantly, the URIs in subject, predicate, or object position of the template statements can be declared placeholders with the class UriPlaceholder, and similarly for literals with LiteralPlaceholder. Such placeholders are represented as input elements, such as text fields or drop-down menus, in the form that is generated from the template. Currently supported more specific placeholder types include TrustyUriPlaceholder, which requires a trusty URI (such as a nanopublication URI), and RestrictedChoicePlaceholder, which leads to a drop-down menu with the possible options defined by the property possibleValue. For URI placeholders, prefixes can be defined with hasPrefix and regex restrictions with hasRegex, as can be seen in the example above.

Once the user filled in a form that was generated from a template and clicks on “Publish”, Nanobench creates the assertion graph of a new nanopublication by following the template and replacing all the placeholders with the user’s input. For the provenance graph, only a simple prov:wasAttributedTo link to the user’s identifier is currently added (we are working on extending the coverage of templates to the provenance and publication info graphs). In the publication info graph, Nanobench adds a timestamp, specifies the user as the creator of the nanopublication, and adds a wasCreatedFromTemplate link that points to the underlying template nanopublication. Then, Nanobench adds a digital signature element to the publication info graph with a signature made from the user’s local private key, transforms the whole nanopublication into its final state with a trusty URI, and finally publishes it to the server network with a simple HTTP POST request. Within a few minutes or less, it then appears in the user’s feed.

Nanobench currently makes use of the redundancy of the nanopublication services in a very simple way: For each query, it randomly selects two grlc service instances and sends the same query to both. It then processes the result as soon as it gets the first answer and discards the second, thereby increasing the chance of success and lowering the average waiting time. More sophisticated versions of this protocol are of course easily imaginable and will be investigated in future work.

Performance evaluation

In order to evaluate our approach, we introduce here a performance evaluation that we ran on the network of nanopublication services. In the next section we will then look into whether these services are useful to potential end users with a usability evaluation on Nanobench.

Performance evaluation design

For this performance evaluation we wanted to find out how well the two types of services—LDF-based and grlc-based—perform in our network of services, how they compare, and to what extent they are really complementary. For this purpose, we defined a set of concrete queries that we can then submit to both services. We started with the 14 query templates of the grlc-based service, and instantiated each of them with a simple set of parameters to make 14 concrete executable queries. As parameter values, we chose generic yet realistically useful examples that return non-trivial answer sets for the kind of nanopublications that the current templates describe: (1) find_nanopubs restricted to the month 2020-02; (2) find_nanopubs_with_pattern with the predicate value set to foaf:knows; (3) find_nanopubs_with_text on the free-text keyword “john”; (4) find_nanopubs_with_uri to search for nanopublications mentioning a given ORCID identifier; (5–8) of the form find_signed_nanopubs_* are given the same parameters as (1–4); (9) get_all_indexes and (10) get_all_users do not need parameters; (11) get_backlinks and (12) get_deep_backlinks are given the URI of a specific nanopublication, which has a substantial number of backlinks; (13) get_latest_version is given the URI of the first version of a template nanopublication that has afterwards been updated four times; and (14) get_nanopub_count is, like (1), restricted to the month 2020-02.

We can run these queries via the grlc-powered API but we can also use an LDF engine like Comunica to run them against our LDF-based services. The latter comes with some caveats, as the free text queries of find_nanopubs_with_text and find_signed_nanopubs_with_text depend on implementation-dependent non-standard extensions of SPARQL that do not work with LDF-style query execution. Moreover, Comunica currently lacks support for complex property paths, which are needed for get_deep_backlinks and get_latest_version. Queries (3), (7), (12), and (13) can therefore only be run on the grlc-based services but not on the LDF-based ones.

However, the power of the LDF-based services is of course that they can (potentially) run arbitrary SPARQL queries (with some restrictions, as mentioned above). To demonstrate and test this ability, we created another query (15) that in a simple way combines the outputs of two of the currently available templates. Specifically, it checks for a given user (below abbreviated as me:) who he has declared to know via the foaf:knows template, and then searches for papers these people declared to have read via a different template. Thereby, query (15) returns a list of all papers that friends of the user me: have read:

select ?person ?paper where {

me: foaf:knows ?person .

?person pc:hasRead ?paper .

}

This query can be considered a quick-and-dirty solution for exploration purposes, as it misses a number of checks. It does not check that both triples are in the assertion graphs of signed nanopublications, that the first is signed with the public key corresponding to the user in subject position, and that neither of the nanopublications is superseded or retracted. We therefore define query (16) that includes all these checks. This query is more complicated, and we show here for illustration just the SPARQL fragment of the part necessary to check that the second nanopublication ?np2 with public key ?pubkey2 was not retracted:

filter not exists {

graph npa:graph { ?retraction npa:hasHeadGraph ?rh .

?retraction npa:hasValidSignatureForPublicKey ?pubkey2 . }

graph ?rh { ?retraction np:hasAssertion ?ra . }

graph ?ra { ?somebody npx:retracts ?np2 . }

}

The inconvenience of writing such rather complicated queries can be addressed by future versions of the services, which could include predefined options to restrict the query to the assertion graphs and to up-to-date content. The full set of used queries and further details can be found in the Supplemental Material online (DOI 10.5281/zenodo.3994068).

To evaluate the performance of the nanopublication services, we accessed them in a clearly defined setting from a number of different locations from personal computers via home networks, by running the 16 queries specified above on all service instances of both kinds. For that, we created a Docker image that accesses the grlc-based services with simple HTTP requests via curl and the LDF-based ones with the Comunica (https://github.com/comunica/comunica) engine 1.12.1. The results as well as the execution time of all the calls are recorded, which is then used to evaluate the performance. For both kinds of services, the timeout is set to 60 s.

Performance evaluation results

We ran the Dockerized evaluation process described above at five places in four different countries. Each of them ran all of the compatible queries on each of the six existing service instance for both of the two kinds. For each query we therefore have 30 outcomes for grlc and another 30 outcomes for LDF. These outcomes fall into the general categories of timeout, error, and full result. In the case of the LDF-based services, timeout and error outcomes can come with partial results. Figure 4 shows a summary of these overall outcomes.

Figure 4 Overall outcomes per query and kind of service, executed from five locations.

With grlc, 96% of the calls succeeded and only 4% resulted in an error (mostly due to downtime of one particular service). With LDF, 73% fully succeeded, 21% reached the timeout, and 6% gave an error. The latter two sometimes gave partial results: overall 6% reached a timeout while still giving partial results, and overall 3% gave an error with a partial result. For LDF, these types of outcomes are not evenly distributed. Two queries—find_nanopubs_with_uri (4) and get_all_indexes (9)—never fully succeeded, but the former sometimes gave partial results. For the remaining queries, however, these LDF calls returned at least a partial result in 97% of the cases. Except for query (1) in addition to the above mentioned (4) and (9), the full result was always received from at least one of the servers in LDF mode. For grlc, this was the case for all queries. A client checking multiple servers would therefore have eventually received the full result. For query (1) in LDF mode, this was true for 4 cases out of 5.

Next, we can look at the time performance. Table 1 shows the average execution times per query and service type, including only the calls that returned a full result. The successful queries to the grlc services took on average from 0.21 to 6.46 s. For the LDF services, these numbers range from 1.53 to 35.26 s (but they can be a bit misleading as they ignore the fact that the LDF services repeatedly hit the time limit of 60 s). For the queries that could successfully be run on both kinds of services, LDF is on average 7.18 to 86.50 times slower than grlc.

Table 1 Average execution times of the successful query executions in seconds.

	Query	grlc	LDF	L/g	
1	find_nanopubs	1.02	35.26	34.48	
2	find_nanopubs_with_pattern	0.55	6.69	12.20	
3	find_nanopubs_with_text	6.46			
4	find_nanopubs_with_uri	0.78			
5	find_signed_nanopubs	0.49	20.77	42.05	
6	find_signed_nanopubs_with_pattern	0.73	9.57	13.04	
7	find_signed_nanopubs_with_text	1.54			
8	find_signed_nanopubs_with_uri	0.34	29.53	86.50	
9	get_all_indexes	3.52			
10	get_all_users	0.65	31.09	47.71	
11	get_backlinks	0.21	1.53	7.18	
12	get_deep_backlinks	0.68			
13	get_latest_version	0.71			
14	get_nanopub_count	0.23	6.54	28.29	
15	papers		2.30		
16	papers_x		10.07		

Importantly, the queries that do not follow a predefined pattern (15) and (16) gave the full result with LDF in 97% of the cases and ran reasonably fast. The quick-and-dirty version (15) required on average 2.30 s, whereas the thorough one (16) completed on average after 10.07 s.

Usability evaluation

Now that we know that the services perform reasonably well, we wanted to find out whether this general approach and our specific Nanobench tool indeed makes it easy for users who might not be experts in Linked Data to publish their own small data entries.

Usability evaluation design

We wanted to test the usability of Nanobench in a real setting, where users actually publish nanopublications. For that we wrote detailed instructions on how to install and use Nanobench and its publication feature, which includes downloading the latest Nanobench release, running it locally, accessing Nanobench through their web browser, completing the Nanobench profile, accessing the list of templates, and finally filling in and submitting the publication form generated from a chosen template. Through mailing lists, social media, and personal contacts, we tried to convince as many people as possible to try out Nanobench and to publish some nanopublications on their own.

Next, we created an anonymous usability questionnaire, consisting of the ten standard questions of the widely used System Usability Scale (SUS) (Brooke, 1996). We added to that the questions “Have you published RDF/Linked Data before?” and “Have you digitally signed RDF/Linked Data before?”, and as a follow up to each of them—if the answer was “yes”—whether Nanobench was harder or easier to use for publishing and signing Linked Data, respectively, compared to how they previously did it. The responses were on a 5-point Likert scale from 1 (Nanobench was harder) to 5 (Nanobench was easier).

We sent this questionnaire to all the Nanobench users who published at least one nanopublication (not counting their introduction nanopublication), excluding the co-authors of this paper and their close relatives. Further details, including instructions and questionnaire, can be found in the supplemental material online (DOI 10.5281/zenodo.3994066).

Usability evaluation results

Overall, 42 users registered in the decentralized system by publishing an introduction nanopublication. A total of 29 of them (69%) also linked this introduction nanopublication from their ORCID accounts, which was a step that was marked as optional. Collectively, they published 81 nanopublications, not counting their introduction nanopublications, via the use of seven distinct templates. After applying the exclusion criteria defined above, we arrived at a set of 29 users to whom we sent the anonymous usability questionnaire (this set of users is overlapping but different from the set of 29 users mentioned just above). After sending up to two reminders, we received responses from all of them.

On the question of whether they had published Linked Data before, 21 respondents (72%) said they did. 20 of them (95%) reported that Nanobench was easier to use compared to how they previously published Linked Data, with the remaining one being indifferent (score of 3). The average was 4.5 on the 5-point Likert scale. Of the 21 respondents, only three (14%) stated that they had previously digitally signed Linked Data. All three of them found Nanobench easier, giving two times a 5 and once a 4 as responses (average 4.7).

Table 2 shows the results of the SUS questions. Overall, our system achieved a SUS score of 77.76, which is clearly above the average score reported in the literature (70.14) and is roughly in the middle between “good” and “excellent” on an adjective scale (Bangor, Kortum & Miller, 2008). Interestingly, if we only consider the eight respondents who stated they had never published Linked Data before, this value is even better at 85.94, clearly in the “excellent” range.

Table 2 SUS usability evaluation results.

		better →		
SUS questions:	odd questions:	1	2	3	4	5		
	even questions:	5	4	3	2	1	Score	
1: I think that I would like to use this system frequently	0	3	9	13	4	65.52	
2: I found the system unnecessarily complex.	0	0	3	11	15	85.34	
3: I thought the system was easy to use	0	1	1	13	14	84.48	
4: I think that I would need the support of a technical person to be able to use this system	1	2	5	7	14	76.72	
5: I found the various functions in this system were well integrated	0	1	7	14	7	73.28	
6: I thought there was too much inconsistency in this system	0	1	2	15	11	81.03	
7: I would imagine that most people would learn to use this system very quickly	0	3	6	14	6	69.83	
8: I found the system very cumbersome to use.	0	0	1	17	11	83.62	
9: I felt very confident using the system	0	1	6	15	7	74.14	
10: I needed to learn a lot of things before I could get going with this system	0	1	4	8	16	83.62	
	Total:	1	13	44	127	105	77.76	

The participants were moreover given the possibility to provide further feedback in a free-text field. We received a variety of comments for further improvement, but except for the point that the required local installation was somewhat inconvenient, no point was mentioned more than once. The other comments concerned the search page being confusing (this part of the interface was indeed not the focus of the study), the lack of support for batch publishing of multiple similar nanopublications, the lack of integrated ORCID lookup, the relatively small number of general-purpose templates, the lack of RDF prefix recognition, the fact that not all lengthy URIs are masked with readable labels in the user interface, and the fact that the confirmation checkbox did not mention the possibility of retraction. A further comment was that a command-line interface would have been preferred in the particular context of the given participant. Such a command-line interface actually exists (as part of the nanopub-java library; Kuhn, 2016) but was not the focus of this study.

Discussion and conclusion

The results of the performance study described above confirm that the tested kinds of queries can be efficiently answered by at least one of the two types of services, and that these two service types are indeed complementary. The grlc services run reliably and fast on the types of queries they are designed for. The LDF services can run most of these kinds of queries too, albeit in a much slower fashion, and they are reasonably fast for simple kinds of unrestricted queries. The results of the usability study indicate that our Nanobench client application connecting to these services is indeed easily and efficiently usable, even for users with no prior experience in Linked Data publishing.

In future work, we are planning to improve a number of aspects of the involved tools and methods. For example, our approach does not yet exploit the full potential of replication in our decentralized setting. Existing work has shown that a client-side algorithm can enable effective load-balancing over TPF servers (Minier et al., 2018), and we plan to extend this work to QPF. As another example, our otherwise decentralized approach currently uses centralized ORCID identifiers. We are therefore investigating decentralized forms of authentication, such as WebID-OIDC (https://github.com/solid/webid-oidc-spec) or an approach similar to the web of trust (Caronni, 2000), where public keys are found based on personal trust relationships that could themselves be published as nanopublications.

Such a web of trust could then also allow users in the future to find trustworthy services. This could include meta services whose task is to monitor and test other kinds of services, so clients could make an informed decision on which service instances to rely on. This is currently difficult, as there is no guarantee that all services are well-behaved and return complete and correct results. Clients could already now deal with this by taking random samples of nanopublications from the publishing servers and check whether the query services correctly return them, but this is quite resource intensive.

Another issue that needs to be taken care of in future work is identity management when private keys are compromised, lost, or simply replaced as a measure of precaution. For that, we envisage that introduction nanopublications are extended so users can also list old public keys. On top of that, we are going to need a method for users to re-claim old nanopublications they signed with an old key that has since been compromised by a third party (possibly by linking to them with an index nanopublication signed with a new key). This will also require modifications in how we deal with retracted and superseded nanopublications, as they might then be signed with a different key. This is not trivial but can be dealt with within our framework, as opposed to Blockchain-based solutions where identity is inseparably linked to private key access. Currently, users need to install Nanobench locally to ensure secure private key access and proper decentralization, but a more flexible and more powerful handling of private keys as explained above will also allow us to provide login-based public Nanobench instances with their own sets of private keys, which in turn can significantly increase the ease of use of our approach.

More work is also needed on the templating features to also cover the provenance and publication info graphs. We also plan to more closely align our templating vocabulary with existing RDF shape standards. Moreover, we are working on making our templating approach more general and more powerful, by adding repeatable statement patterns among other features, such that we can express, for example, templates of templates and thereby allow users to create and publish their own templates directly via Nanobench.

The tools and applications we described above in a sense just scratch the surface of what can become possible with our general approach in the nearer future, from Linked Data publications of the latest scientific findings, to formally organized argumentation and automated real-time aggregations. We believe that our approach of semantic micro-contributions could in fact be the starting point of bringing Linked Data publishing to the masses.

Supplemental Information

Supplemental Information 1 The repositories for the performance and usability studies.

Contains the code and data that was used and generated for the performance evaluation and the usability study.

Click here for additional data file.

Additional Information and Declarations

Competing Interests

Author Contributions

Data Availability

1 For simplicity, URLs for p and g are prefixed, whereas they will be expanded in practise.

2 A live example of a QPF client that can query over this API can be found at http://query.linkeddatafragments.org/#datasources=https%3A%2F%2Fldf.nanopubs.knows.idlab.ugent.be%2Fnp.

Haris Antonatos is employed by SciFY. The authors declare that they have no competing interests.

Tobias Kuhn conceived and designed the experiments, performed the experiments, analyzed the data, performed the computation work, prepared figures and/or tables, authored or reviewed drafts of the paper, and approved the final draft.

Ruben Taelman performed the experiments, prepared figures and/or tables, authored or reviewed drafts of the paper, and approved the final draft.

Vincent Emonet performed the experiments, authored or reviewed drafts of the paper, and approved the final draft.

Haris Antonatos performed the experiments, authored or reviewed drafts of the paper, and approved the final draft.

Stian Soiland-Reyes performed the experiments, authored or reviewed drafts of the paper, and approved the final draft.

Michel Dumontier performed the experiments, authored or reviewed drafts of the paper, and approved the final draft.

The following information was supplied regarding data availability:

Supplemental data for the performance evaluation is available at Zenodo:

Tobias Kuhn, & Vincent Emonet. (2020, August 21). peta-pico/nanopub-services-eval 1.0 (Version 1.0). Zenodo. DOI 10.5281/zenodo.3994068.

Supplemental data for the usability evaluation is available at Zenodo:

Tobias Kuhn. (2020, August 21). peta-pico/nanobench-usability-eval 1.0 (Version 1.0). Zenodo. DOI 10.5281/zenodo.3994066.

The code for Nanobench (release nanobench-1.7) is available at Zenodo: Tobias Kuhn, & Vincent Emonet. (2020, November 26). peta-pico/nanobench: nanobench-1.7 (Version nanobench-1.7). Zenodo. DOI 10.5281/zenodo.4292171.

The code for the nanopublication services (release nanopub-services-1.0) is also available at Zenodo:

Tobias Kuhn. (2020, November 26). peta-pico/nanopub-services: nanopub-services-1.0 (Version nanopub-services-1.0). Zenodo. DOI 10.5281/zenodo.4291594.

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
