# Peer review of "Semantic micro-contributions with decentralized nanopublication services"

_PeerJ Computer Science, doi:10.7717/peerj-cs.387_

## Round 0.1 · original submission · Major Revisions

We have now received reviews on your submission. All three reviews suggest "major revision" with several concerns that need to be addressed especially concerning the exact contribution of the paper and the novelty of your work with regard to the state of the art contributions. Thus the submitted version cannot be accepted and a major revision is recommended.

If you decide to submit a revised version, please do tackle the following issues:

- an explicit comparison with your previous work needs to be added (See R1 and R2) and an explicit comparison in terms of research questions to be answered in this specific work with regard to previous work needs to be clarified. I suggest the authors seriously address the concern of what is the contribution of this paper in the context of previous work and also to consider whether to present their contribution by explicitly adding research questions as they would address also the comment of R3 on the scientific vs technical contribution of your paper.

- a discussion on the issue of trust and delegation of integrity constraints to the server (see R1)

- an improved discussion on scalability even if only for future works so as to address the issue somehow (see R1)

- a discussion of the novelty wrt other interfaces for collecting semantic knowledge and decentralised semantic knowledge (see R3)

- a consolidation of the evaluation as pointed out by R2 and R3.

·

Basic reporting

The paper presents exensions of the nanopublication architecture / server network previously published by some of the authors (Kuhn et al, 2016), with the overall aim to provide non/expert end users with an effective and easy to use solution to publish authenticated nanopublications - so-called "semantic microblogging". More in details, newly contributed extensions w.r.t. (Kuhn et al, 2016) consist in the support for:
* digitally signing nanopublications (section 3.1);
* superseding/retracting nanopublications (section 3.1);
* querying nanopublications via LDF (via new quad pattern fragments extension - section 3.2) and grlc APIs (sections 3.3);
* accessing these APIs and manipulating signed nanopublications via the new Nanobench web application via a template mechanism (section 3.4);
The new features are assessed in terms of performance and usability. Evaluation code and data is made available in supplemental material.

The manuscript report on an incremental work w.r.t. (Kuhn et al, 2016). The novel contributions appear appropriate - in terms of significance and scope - for a PeerJ publication according to the Journal reviewing guidelines.

Background and references are overall adequate, and the paper is mostly self-contained with two minor exceptions:
* the paper should provide an explicit discussion of how this work relates to (Kuhn et al, 2016) including which additional problem(s) / research question(s) it addresses and which novel contributions it provides, as currently I had to go back and read the other paper to figure out these aspects;
* the paper (section 3.2) should provide more details about how triple pattern fragments have been extended to quad pattern fragments, as interested readers currently have to figure out this information in the linked specification.

The paper is overall well written and structured, the quality of English is good, and listings, figures and tables are appropriate. Some suggestions:
* [section 1] I suggest providing more details on the contributions of the paper, possibly reusing content from [section 3, lines 106-114]
* [section 1, line 33] "Such files are not regularly indexed ... and which therefore often lack discoverability" -> typo: remove "which"
* [section 3.1, line 159] "that can work on any URL scheme" -> suggest writing "any URL scheme and identification mechanism providing dereferenceable user identifiers"

Experimental design

A research question is not explicitly stated in the paper (if strictly required for the journal), although the paper provides new useful features / services extending the nanopublication architecture of (Kuhn et al, 2016) and therefore mostly addresses the same research question of that work:

[[
Can we create a decentralized, reliable, and trustworthy system for publishing, retrieving, and archiving Linked Data in the form of sets of nanopublications based on existing web standards and infrastructure
]]

with a specific focus on efficacy and usability by non-expert end users. In any case, I would suggest authors to improve the introduction and be more specific about the particular problem addressed here and how this work fits in / extends the one presented in (Kuhn et al, 2016).

I have some concerns with the approach design in terms of trust assumptions, scalability and fault tolerance, which I report in the general comments section. Ignoring these concerns and leaving out scalability and fault tolerance from evaluation dimensions, then I believe the experimental design is appropriate.

Approach code and evaluation code and data are provided (via supplemental material / external links), this way ensuring adequate reproducibility.

Validity of the findings

The findings drawn from the two evaluations (section 6) are overall justified by the evaluation results (sections 4.2, 5.2).

Concerning usability, I have no remarks except suggesting - if possible, i.e., if Nanobench logic is fully implemented browser-side - to deploy Nanobench on the server network so that users don't have to install it locally, this way further enhancing usability. If that is possible and there were user complaints about the tool installation, then authors may argument / roughly quantify in the paper how usability results may improve if installation steps (and possibly associated complaints) are removed.

Concerning performance, I'm a bit puzzled by the success/error percentages and analysis of section 4.2 / figure 3. Put that way, it would seem that there is no guaranteed way to successfully query the two APIs (i.e., it's expected to encounter errors!), although multiple servers are available and a client (like Nanobench) is in principle able to use different servers and thus cope with a specific server error/timeout (as reported for grlc). I would suggest authors to extend the analysis by considering also a different notion of success / error as perceived by the end user: a query is successful for a certain client location if there is at least a server able to answer it without errors (so a client contacting different servers from that location would eventually succeed). Put this way, I think grlc queries will likely achieve 100% success rate, and LDF queries would also improve. Concerning the latter, it would be interesting to consider also (client-side) timeouts larger than 60s.

Additional comments

Major comments:

* System design and assumption of trusting the server network -- the proposed solution featuring the two grlc and LDF APIs assume implicit trust in the servers offering these APIs. For instance, we must assume that they will return correct and complete (pending timeouts/errors) answers to queries, that the data they provide match the nanopublication data uploded by users, and that data in the two administration graphs (if publicly accessible, as it seems) is correct and complete. Note that these trust assumptions, which are relevant in this grlc+LDF service layer and likely in any other service layer, are less relevant for the core nanopublication distributed architecture, since there the client can only lookup a trusty URI and then it can verify the integrity of the associated content itself. Now I'm not much concerned with these trust issues (although it would be nice to discuss them in the paper). My point, instead, is that since a client already has to trust the servers, then why not delegate all integrity checks to the servers, instead of asking clients to perform them themselves through complex queries such as Query 16 (section 4.1, supplemental material) that negatively affect usability and performance? For instance, the server may offer a separate grlc/LDF endpoint offering only the latest versions of nanopublications whose signature and integrity have been successfully checked, this way greatly easing performance and usability (i.e., Query 15 will suffice). If trust in the server network cannot be assumed, then a possibly interesting solution would be to offer an API that, given a query method (grlc, LDF, or other), returns only / also the raw content of the nanopublications from which query results can be extracted from, so that the client can verify the authenticity / integrity of such data and compute / verify the query results itself. If I got this aspect wrong, or if there are precise motivations for designing the system this way, I suggest authors to comment or argument in the paper.

* Scalability and fault tolerance -- these are relevant aspect of a distributed system like the one being proposed, especially given sentences like the one concluding section 6 (semantic microblogging possibly being "the starting point of bringing Linked Data publishing to the masses"), but also in case the scope of the proposed solution is not web-wide but restricted to specific use cases / deployments as argued in (Kuhn et al, 2016). Concerning scalability, while I trust that the key-value stores used by the core nanopublication architecture are scalable, I have some concern about the scalability of the triplestore backing the new APIs since the system design asks for storing all queryable nanopublication data in it. Given the breadth of the paper contributions, I'm fine if scalability is left out, but I would suggest explicitly stating it and possibly pointing at more efficient designs (as future work) in case the authors already have ideas to that respect. Concerning fault-tolerance, the experimental setting using Nanobench already permits to roughly assess it (see suggested analysis with different error/success notion), but also here I think authors may present the solution as an initial step and explicitly leave out fault tolerance from the paper.


Minor comments:

* [section 3.1, lines 151-152] Concerning the requirement that updates and retractions must use the same private/public key pair: what if such key pair is invalidated and a new key pair is associated to the user? I'm in no way a security expert but I believe a similar issue occurs with legally-recognized digital signatures (where a personal key pair may also expire), so probably similar solutions may be adopted in this case, also to guarantee that previously posted nanopublications are still recognized as legit in case the user later updates his/her key pair.

* [section 3.4, lines 258-260] Editing the ORCID profile to link it to the nanopublication providing the public key appears crucial to me (vs. only optional), as without this step no third party can fully trust the nanopublications issued by a user.

* [section 3.4, lines 291-293] I think what needs to be sorted in the form are placeholders, rather than statements. How does "statementOrder" work precisely, to that respect? Suggest specifying better in the paper.

* [section 3.4] Suggestion: while I acknowledge the usefulness of templates, I think the Nanobench application may benefit from a general RDF file upload functionality allowing users to post arbitrary content not supported by available templates (size limits and data validation may be enforced prior to publication, in this case).


Overall, I believe the work described in the paper is interesting but the paper can be further clarified / improved, so I recommend a major revision and another review round.

·

Basic reporting

The paper tackles the very relevant issue of authoring and publishing Linked Data. It describes a decentralized approach, based on nanopublications and inter-connected services, that allows users to publish small sets of statements. The system incluses a template-based interface, called Nanobench, that helps authors to compose these statements. The article is completed with an empirical evaluation of both the performance of the architecture in answering queries and the usability of the interface.

I found the work interesting and solid. On the other hand, I would suggest a major revision mainly to improve the presentation.
More comments in the 'Experimental design' review.

Experimental design

I see two main issues. First of all, the paper does not make a clear distinction between the specific contribution of this work and the overall approach based on nanopublications proposed by the same authors. The background section points to some previous works but the paper should better state which research questions are answered by this specific work and which is the advancement to the state-of-the-art.

Related to this, the paper combines two very different perspectives and the connection between these looks rather weak. On one side, there are architectural aspects (identifying and distributing statements, building services, querying, etc.); on the other, usability ones (template-based authoring of LD). The paper, and in particular the introduction, should identify the exact focus and novelty. Is it on the interface or on the services? Or both? How is different from other contributions?
This is also reflected on the evaluation part. The two parts are quite disconnected and the usability one looks weaker.
If the focus is on the semantic blogging approach, the evaluation should involve more users and go deeper in the features of the interface.

The paper would also benefit a stronger separation between the literature review, to identify the limitations of the existing approaches, and the background section, to better understand the global approach proposed by the authors.

Validity of the findings

No comment.
Notes about impact and novelty in the 'Experimental Design' review.

Additional comments

Some more detailed comments (including some minors):

- L.17: the sentence “It also imposes a large overhead… …or original content” is not clear in the abstract, please rephrase
- L.18: use “do not” instead of the contract form “don’t”
- L.29: the reference about the remarkable adoption of Linked Data goes back to 2011, should be changed with a more recent one
- L.37: the introduction should also mention other research efforts on decentralized services for publishing data, as described in Section 2
- L.42: the introduction mentions semantic microblogging but this is lost in the rest of the paper; it should be given more relevance; a more detailed comparison between the use of ‘semantic microblogging’ in this work and in (Passant et al. 2008) would also be helpful
- L.53: the relation of ‘semantic publishing’ is not clear here; the introduction does not mention it and it is focused on Linked Data in general; later in the section it becomes clear though
- L.93-103: the discussion about decentralized approaches is too short. The analysis of the reference (Third and Domingue, 2017), for instance, is only a few lines even if this work is very related. A more accurate discussion of the literature would strngthen the overall approach and would help to highlight the novelty of the paper.
- L.106: a picture summarizing the overall approach and components would be very helpful for the readers
- L.121: the proposed approached seems to be weak, since the nanopublication can be signed by anybody claiming any ORCID. This is clarified later in the paper (L.153-) but should be anticipated here
- L.161: QPF is explicity mentioned here for the first time; it should be given more relevance in the first part
- L.192-211: a table showing the queries instead of a long paragraph would be much more readable
- L.212-214: the discussion is very short even if these problems are complex; this part should be extended
- L.254: a list of predefined templates is loaded; is it possible to add new templates? How? Is it complex? Did authors evaluate this part?
- L.327: add some details about the process for selecting these 14 queries
- L.419: add details about the test: how many nanopublications were published? tow complex are they? Even if there is the supplementary material (which is very good) some more hints would help
- L.425: if possible, the authors should add some more information about the background of the testers. How did they publish Linked Data in the past? When they say that ‘Nanobench was easier to use’ which other approaches they compared?
- L.435: did testers also provide suggestions to improve the interface?

Reviewer 3 ·

Basic reporting

The paper discusses a novel decentralized strategy for supporting the publishing of semantic microblogs (as defined within the text) into the Linked Open Data (LOD) cloud in a more easy and efficient way.
Beyond the theoretical aspects, the authors presents a user interface through which the users can manually provide new knowledge.
The paper is sound with the scope of the journal, well presented and motivated, and the authors provided the supplementary material for better checking the evaluation.

On the one hand the paper is interesting since the problem of managing all the knowledge of the LOD cloud is known to the community and, at the time of this review, it seems that no specific efforts are planned to address this problem.
On the other hand it seems anyway that the problem is more technical-wise than research-wise. Hence this consideration affects part of the motivation behind your work.
How can the authors comment about this?

Another point is related to two novelty aspects.
First, interfaces and tools for simplifying the gathering of knowledge from users and domain experts have been widely studied in the last decade and several state of the art tools are currently available (MoKi, WebVOWL, ecc.). Is Nanobench of the same family of these tools or it has different aims? Obviously, the fact that Nanobench works with microbloggings while the other tools have been thought for ontology design does not count.
Second, the decentralization of information, and also of semantic knowledge, is something that already started to be investigated. As example consider this paper:
- Harm Delva, Julián Andrés Rojas Meléndez, Pieter Colpaert, Ruben Verborgh: Decentralized Publication and Consumption of Transfer Footpaths. SEM4TRA-AMAR@SEMANTICS 2019
Please argue if your work belongs to this research direction or not and why.

Finally, my opinion the evaluation is incomplete.
How were the queries chosen? Were they validated in some way by some experts or not?
How were the Linked Data Fragment built?
Here the problem is quite thorny since there are many way for creating the fragments and also the hardware equipment significantly affect the performance.
Please argue on that.

Experimental design

The experiments have been conducted properly even if further details and discussions should be provided as highlighted in the report above.

Validity of the findings

No comment.

Additional comments

Overall, the work presented is good and I do not find big reasons for rejection.
However, I warmly invite the authors to address the comments in order to improve the quality of the paper and to make it more of impact for the community.

---

## Round 0.2 · Minor Revisions

After a careful evaluation I concur with Reviewers 1 and 2 in indicating that the paper is worth publishing in PeerJ Computer Science. Its content (showing the usefulness of a technological approach) has the typical scientific value of good ''in use'' papers which fit in the scope of the journal.

The reviewer(s) have recommended publication, but also suggest some minor revisions to your manuscript. Please carefully consider these comments when revising your paper.

Once again, thank you for submitting your manuscript to PeerJ and I look forward to receiving your revision

Sincerely,
Chiara Ghidini

·

Basic reporting

The manuscript is a revised version of the original submission I previously reviewed, so I will focus my review on authors' changes. I would also like to thank the authors for their response and for having addressed the comments raised in my previous review.

The paper research question, novel contributions, and relation to previous authors' work (esp. Kuhn et al, 2016) are now explicitly stated, positively addressing a concern in my original review.

The paper presentation is good and has further improved in this revision, also due to the above clarifications. I also appreciate the added approach diagram (Figure 1). Spotted typos for authors' convenience:
* [section 3.2, line 202] "allows allow" -> "allow"
* [section 3.2, line 230] "can be join" -> "can be joined"

Experimental design

A research question is now clearly indicated in the introduction, and appears appropriate. I previously found the experimental setup appropriate and appreciated the availability of code and data to ensure reproducibility.

Validity of the findings

As previously reported, I believe the findings summarized in the paper (section 6) are justified by the conducted evaluation and its results (sections 4, 5).

I appreciate the analysis of which / how many queries can be successfully evaluated by trying multiple servers (section 4.2), which addresses a concern I previously raised and demonstrates the benefit of replication.

I thank the authors for their clarification (in the authors' answer) about the difficulties of deploying Nanobench on the server network. While I see technical solutions to the central point of failure issue (e.g., by replicating Nanobench on multiple servers and using techniques for directing users to the closest online server, similar to the ones adopted in content distribution networks), I didn't consider the need for Nanobench to access the user's private key and I find this much more tricky to deal with. I agree with the authors' arguments, and I think it may be worth explicitly mentioning this private key issue when discussing private keys in section 6 of the paper.

Additional comments

The revised manuscript provides indications of how the proposed solution would accommodate future work extension to address concerns related to trust in the infrastructure, scalability, and fault tolerance. I think this is appropriate and sufficient for this paper, whose main contribution is to lay the foundations for enabling end users to easily publish nanopublications in an authenticated and decentralized way.

Related to trust, I previously suggested to further rely on the underlying assumption of trust in the infrastructure (which can be "lessened" as hinted by authors, but not completely eliminated) in order to simplify end user queries, exemplifying this with Query 16, which I found very complex from an end user point of view. I agree with the authors' response and the fact that it must be possible to formulate complex queries such as Query 16 that can access all data, including retracted or non-authenticated one. I just wondered how frequent such queries will be, with respect to "simpler" queries interested in up-to-date and authenticated data only, and therefore advocated for having the trusted servers to provide further data / access mechanisms to simplify those "simpler" queries. I think the solution suggested by authors of adding additional triples in the administrative graph (e.g., to materialize checks precomputed by servers) can be appropriate to that respect, and it may be worth to mention that in that paper when commenting on the complexity of Query 16.

I thank the authors for all their clarifications to the minor comments I made in my previous review. I'm fine with their answers, esp. regarding leaving key invalidation as future work and the statement-based organization of forms justifying the way 'statementOrder' is used (I imagine that even if a placeholder is used in multiple statements, the end user will be asked to fill it only once in the form).

Overall, I believe the paper has improved and all of my concerns have been addressed, in the paper or the response and in some cases (which I agree with) leaving them as future work. The few additions to the paper I'm suggesting in this review are very minor and the manuscript is already fine as it is now, so I recommend acceptance.

·

Basic reporting

The presentation in the revised version was improved, with enough references, contextual information, figures and structured data.

More details in the general comments.

Experimental design

The experiments are clearly illustrated in the revised version, as well as the research and technical questions.

More details in the general comments.

Validity of the findings

Data and code are freely available and the supplementary material is rich.

More details in the general comments.

Additional comments

The revised version of the article looks very clear to me, and the rebuttal was satisfying. The introduction, in particular, explains very well how the article is positioned in the literature and how its different parts are connected each other (which were my main concerns on the first submission). I also like the term micro-contributions. On the same line, this new version clearly explains the relation between this work and the previous ones by the same authors.

In the current form, the article looks ready for publication in my opinion. Yes, it is focused on technical and implementation aspects but I think it is a nice piece of work in the overall framework of nanopublications.

Just some (very) minor points:
- lines 52-54: the paragraph looks a bit convoluted, please check
- lines 183-185: some references to research works on retraction would further complete the article
- lines: 248-251: add a very brief description on what can be done with each service; the text mentions how the services are implemented and the list of possible queries is right after but some anticipation would help the reader
- lines 505-508: if I understood correctly from the rebuttal, the authors do not have information about the background (role, position, background knowledge, mastered tools, etc.) of the testers. Is that true? If you have any details, it would be helpful to include them in the paper
- lines 508-: I would include the suggestions of the authors about the Nanobench interface, even if they were given only by a few testers, just to complete the picture

Reviewer 3 ·

Basic reporting

No comment.

Experimental design

No comment.

Validity of the findings

No comment.

Additional comments

I want to thank the authors for the answers provided.
The changes carried out on the paper satisfy the issues highlighted on the first version of the manuscript.
The only doubt that I still have is the research contribution of this work.
Actually, I do not have anything against papers having a stronger technical contribution rather than a research one.
I only want to highlight this aspect to the journal's Editors in order to judge the paper properly with respect to the journal's policies.

---

## Round 0.3 · accepted · Accept

Dear Tobias, Ruben, Vincent, Haris, Stian, and Michel

Thank you for this revision and for explaining your changes based on the reviewers’ comments. Everything has been addressed properly. Congratulations!

Thank you again for having selected PeerJ Computer Science!

Best regards,
Chiara Ghidini